# Evolving Deep Neural Network's Weights at ImageNet Scale

## Abstract

Building upon evolutionary theory, this work proposes a deep neural network optimization framework based on evolutionary algorithms to enhance existing pre-trained models, usually trained by backpropagation (BP). Specifically, we consider a pre-trained model to generate an initial population of deep neural networks (DNNs) using BP with distinct hyper-parameters, and subsequently simulate the evolutionary process of DNNs. Moreover, we enhance the evolutionary process, by developing an adaptive differential evolution (DE) algorithm, SA-SHADE-tri-ensin, which integrates the strengths of two DE algorithms, SADE and SHADE, with trigonometric mutation and sinusoidal change of mutation rate. Compared to existing work (e.g., ensembling, weight averaging and evolution inspired techniques), the proposed method better enhanced existing pre-trained deep neural network models (e.g., ResNet variants) on large-scale ImageNet. Our analysis reveals that DE with an adaptive trigonometric mutation strategy yields improved offspring with higher success rates and the importance of diversity in the parent population. Hence, the underlying mechanism is worth further investigation and has implications for developing advanced neuro-evolutionary optimizers.

## 1 Introduction

Deep neural networks (DNN) (Krizhevsky et al., 2009; LeCun et al., 2015) have seen remarkable advancements, leading to exceptional performance across a broad spectrum of learning tasks and applications, such as visual tasks (Chen et al., 2020) and natural language processing (Stiennon et al., 2020). Artificial Neural Networks (ANNs) are modeled on the structure and function of the interconnected neurons in the human brain. Training ANNs are equivalent to the search of network weights that optimize a desired loss function, in which intricate network architectures determine the high-dimensional function space. Back-propagation (BP) and its variants (e.g., ADAM (Kingma & Ba, 2014)) have established themselves as the most widely used, thanks to their ability to explicitly utilize the gradients of the loss function and enable the training of extremely deep neural networks. Nonetheless, BP has several weaknesses (Gong et al., 2020). For example, it requires differentiable loss functions and suffers from the sensitivity to hyperparameters, the problems of vanishing or exploding gradients, slow convergence, and high computational requirements. Recently, a forward-forward algorithm that does not require BP for training DNN has been proposed by Hinton, and we envision the advancement of the line of research (Hinton, 2022). One of the feed-forward approaches, for training DNN without BP, is meta-heuristic approaches, for instance, evolutionary algorithms (EA). EAs have successfully evolved a population of solutions to a plethora of complex optimization problems (e.g., non-convex and NP-hard problems) and real-world problems where traditional methods fail. Utilizing EAs to evolve DNN architectures and weights is termed neuro-evolution, which bridges DNN optimization and evolutionary theory for algorithmic development, interpretation, and analysis. An interesting analogy between neural network architecture, dataset, species, and environment are discussed in Sec. S1 of the Supplementary Materials (sup).

Here, we re-investigate the integration of BP-based and EA-based methods, as illustrated in Fig. 1). The proposed work differs from other EA-based methods that train ANNs from random initialization, which imitates the primordial soup (i.e., prior to the formation of the first species, primordial ancestor) in Primordial soup theory (Taylor, 2005). The initialization that implements primordial soup can be the reason for the slow convergence due to the unmet conditions for the evolutionary starting point. Inspired by Hinton's work of pretraining the neural network using restricted Boltzmann

Figure 1: **The conceptual illustration of Darwinian evolution on DNNs.** A population of pretrained DNNs evolves their weights in the environment specified by datasets and loss function.

machines to avoid the vanishing of gradient, we use the BP-based optimizer (i.e., ADAM) for the pretraining of DNN and EA-based methods to evolve DNN's weights. We consider the pretrained DNNs, obtained from the ending epochs, as the primordial ancestor (i.e., the first species), the starting point of evolution. Subsequently, Darwinian evolution is implemented using differential evolution (DE). BP-based algorithms allow a single neural network to accumulate knowledge and learn representation from data, while neuro-evolution emulates the evolutionary aspects DNNs. The proposed strategy applies to any pretrained models, facilitating the evolution of fitter offspring (i.e., enhanced pre-trained models). The concepts are validated using various computer vision datasets and DNN architectures.

## 2 RELATED WORK

A deep learning model is parameterized by deep neural networks' weights that regulate the strength of the connection between neurons. Typically, the weights are adjusted to optimize a loss function formulated for a learning task. The process of weight adjustment is called DNN training, which can be broadly categorized into gradient descent-based (GD) methods (Kingma & Ba, 2014) and evolution algorithm-based (EA) methods (Stanley et al., 2019), composing two essential ingredients of brain adaptation: learning and evolution (Yao, 1999). Popular GD methods include stochastic gradient descent (SGD) and its variants with momentum and adaptive learning rates: AdaGrad, RMSProp, and Adam (Kingma & Ba, 2014). While GD methods are commonly used, it is known to be local search methods that suffer from a few problems, such as gradient vanishing/exploding, getting trapped at local optima, and over-fitting (Yang et al., 2021). To circumvent the challenges, a few works proposed to combine, average, or ensemble multiple DNN models. Weight averaging (WA) (Izmailov et al., 2018) and model soup (Wortsman et al., 2022) average the neural network weights, while ensembling (Garipov et al., 2018) averages the output of the feature map. These methods are used as the non-EA baselines in this study.

### 2.1 EA-BASED WEIGHT OPTIMIZATION

EA-based methods provide gradient-free ways to DNN training, where a population of neural network topologies and weights evolves for better fitness globally (Stanley et al., 2019). Popular EAs algorithms for optimizing DNN include genetic algorithms (Montana et al., 1989), genetic programming (Suganuma et al., 2017), differential evolution (DE) (Pant et al., 2020), and evolution strategies (Salimans et al., 2017). Parsimonious neural architectures have been designed through neuro-evolution (e.g., NEAT (Stanley & Miikkulainen, 2002)) with enhanced performance. Moreover, neuro-evolution techniques (e.g., evolution strategy (Salimans et al., 2017)) have been demonstrated to achieve better results in reinforcement learning tasks compared to deep Q-learning, the policy gradient algorithm A3C (Mnih et al., 2016) among others. However, EA-based methods were only reported to work well on small datasets and small DNNs (Piotrowski, 2014). When optimizing DNNs' weights on large-scale datasets, EA-based methods suffer from very slow (or failure of) convergence, given a large number of model parameters and a complex search space for obtaining the deep representation. Piotrowski reported the stagnation issues of several variants of adaptive DE, such as SADE, JADE, and DEGL, in optimizing network weights for regression problems (Piotrowski, 2014). Sun et al. proposed efficient gene encoding, utilizing the concepts of null spaces, for DNN on

unsupervised learning tasks at MNIST and CIFAR-10 scales (Sun et al., 2018). However, none of the existing EA-based weight optimization methods demonstrated the scalability to ImageNet.

## 2.2 COMPLEMENTARITY OF EA AND GD

Several works hybridize GD and EA for DNN training for combined strengths. Lehman et al. utilized the output gradients for safe mutations to regulate the degrees of mutation, preventing the functionality of DNN from breaking down due to evolutionary operators (Lehman et al., 2018). Cui et al. combined the GD and EA by alternating between SGD and EA steps for complementary strengths of them (Cui et al., 2018). Yang et al. guided the multiobjective EA's search direction using gradients by designing a gradient-based simulated binary crossover (SBX) operator (Yang et al., 2021). Xue et al. developed an ensemble of DE and ADAM to train MLPs, where the two algorithms evolved two populations of neural network weights (Xue et al., 2022). Nonetheless, the datasets studied are small, and the max number of DNN parameters was only 2555. The generalizability to a larger dataset and deeper DNN was not shown.

## 2.3 EVOLVING PRETRAINED MODELS

One special case of hybridizing GD-EA is using EA to enhance existing pre-trained models trained on large-scale datasets (e.g., ImageNet). The line of research is precious, given the growing numbers of publicly available pre-trained models and large-scale datasets. Gong et al. proposed hybrid cooperative coevolution with BP, where BP is used as the starting point of evolution (Gong et al., 2020). The cooperative coevolution was implemented by decomposing the tasks of adjusting DNN's weights into many subtasks based on the proposed neuron maturity. The decomposition reduced the computational cost while enhancing the performance of DNNs for CIFAR-10. As this work designed its own CNN, whether the proposed method generalizes to existing pre-trained models is unclear. Very recently, Whitaker et al. proposed sparse mutation decompositions (SMD) and ensembles to fine-tune existing pre-trained DNNs for ImageNet (Whitaker & Whitley, 2023). The work also studied the impact of mutation strength and sparsity on network sensitivity and performance and claimed the importance of subspace evolution. The authors reported small improvements (smaller than 0.5%) in accuracy on ImageNet for all pre-trained models studied. However, the evolution was only simulated for one generation and did not consider the benefits of incremental, evolutionary changes. Moreover, the authors only reported the results of the ensemble. Hence, whether the ensemble or the proposed SMD contributed to the improvement is unclear. Both works used decomposition to reduce the dimensionality of the search space and might be more vulnerable to local optima (Yang et al., 2021).

This paper attempts to address the research gaps in the potential of EA for enhancing existing pre-trained models by evolving existing pre-trained models in the original search space using the proposed adaptive DE algorithm, SA-SHADE-tri-ensin. The impacts of evolution on DNN are carefully scrutinized.

## 3 METHOD

The task is to learn a neural network parameterized by $\boldsymbol{\theta}$ with dataset $\mathcal{D}$, where each sample $\boldsymbol{x} \in \mathcal{D}$ is with multi-features and a label $y$. Standard method requires minimizing the loss $\mathcal{L}$, formulating as $\boldsymbol{\theta} = \arg\min_{\boldsymbol{\theta}} \mathcal{L}(\boldsymbol{\theta}; \boldsymbol{x}, y)$. We use the cross entropy and the L2-regularization with the regularization factor $\alpha$, $\mathcal{L} = \sum_{i \in \mathcal{D}} y_i \log(f(\hat{y})_i) + \alpha \|\theta\|_2^2$. Back-propagation with gradient-based methods are almost ubiquitously for such optimization. However, the gradient-based optimizer is a point-based local search approach, which may lead to severe sensitivity to parameter initialization and is possible to get trapped into inferior local optima. Based on that, we provide an advanced framework by augmenting the optimization with the evolution algorithm, where Algorithm 1 is presented to learn such a neural network.

The first stage is the initialization with back-propagation methods. This iterative algorithm starts with a random input and fitness evaluation. For each epoch, gradient-based quantity is calculated, and $\boldsymbol{\theta}$ can be updated until the termination condition is satisfied. This produces a set of candidate solutions, $\Theta = \{\boldsymbol{\theta}_i : i = 1, ..., \mathrm{NP}\}$, where NP is called population size, and $\boldsymbol{\theta}_i = (\theta_{i,1}, \theta_{i,2}, ..., \theta_{i,d})$, $d$ is the size of solution space. We pick the last consecutive NP epoch before the termination of this

stage as the initialized population of the second stage. Moreover, we also investigate other tricks on population initialization in Section 3.3.

The second stage is to simulate the evolution of a population of DNNs, which generates new solutions from the current candidate set. A formal DE algorithm includes two main steps: mutation and recombination. For each generation (epoch), the mutation is performed as, $\boldsymbol{\theta}_i^\star = \boldsymbol{\theta}_{r_1} + F \times (\boldsymbol{\theta}_{r_2} - \boldsymbol{\theta}_{r_3})$, where $i = 1, ..., m$ and $r_1, r_2, r_3$ are random integers less than $m$, different from $i$ and other. $F$ is the scaling factor. The recombination is performed as,

$$\theta_{i,j}^\star = \begin{cases} \theta_{i,j}^\star & \text{if } \text{rand}(0,1) \leq Cr, \\ \theta_{i,j} & \text{otherwise.} \end{cases} , \quad \boldsymbol{\theta}_i = \begin{cases} \boldsymbol{\theta}_i^\star & \mathcal{L}(\boldsymbol{\theta}_i^{\star g}) < \mathcal{L}(\boldsymbol{\theta}_i^g) \\ \boldsymbol{\theta}_i & \text{otherwise} \end{cases} , \tag{1}$$

where crossover is taken place with a pre-set threshold $Cr$, and selection is made on the lowest loss function by direct one-to-one comparison. In this stage, variant differential evolution algorithms are employed, where modified mutation and recombination are depicted in Sec.(3.1) and Sec.(3.2).

---

**Algorithm 1** EALEARNING: learning algorithm. $t$ is the type of networks, and $m$ is the population size. $\delta_1, \delta_2, \Delta_1$ and $\Delta_2$ are pre-set threshold.

---
1: **procedure** EALEARNING($\mathcal{D}$,t,m)
2: $\quad \boldsymbol{\theta} \leftarrow$ RANDOMINPUT$, \Delta f \leftarrow 1, f \leftarrow$ LOSS$(\boldsymbol{\theta}, \mathcal{D}, t)$ $\quad\quad\quad\quad$ ▷ random input and evaluation
3: $\quad$ **while** $f \geq \delta_1 \wedge \Delta f \geq \Delta_1$ **do**
4: $\quad\quad \hat{g} \leftarrow$ BATCHBACKPROP$(\mathcal{D}, t)$ $\quad\quad\quad\quad\quad\quad\quad\quad\quad\quad\quad$ ▷ gradient calculation
5: $\quad\quad \boldsymbol{\theta} \leftarrow$ UPDATE$(\boldsymbol{\theta}, \hat{g})$ $\quad\quad\quad\quad\quad\quad\quad\quad\quad\quad\quad\quad$ ▷ updating parameters
6: $\quad\quad f' \leftarrow$ LOSS$(\boldsymbol{\theta}, \mathcal{D}, t), \Delta f \leftarrow f - f', f \leftarrow f'$
7: $\quad\quad \Theta \leftarrow$ GETS$(\boldsymbol{\theta}, m)$ $\quad\quad\quad\quad\quad\quad\quad\quad\quad\quad\quad\quad\quad$ ▷ store recent m $\boldsymbol{\theta}$
8: $\quad$ **end while**
9: $\quad$ **while** $f \geq \delta_2 \wedge \Delta f \geq \Delta_2$ **do**
10: $\quad\quad \Theta^\star \leftarrow$ MUTATE$(\Theta), \Theta \leftarrow$ RECOMBINATION$(\Theta, \Theta^\star)$ $\quad\quad$ ▷ modified mutation and recombination
11: $\quad\quad$ **for** $\boldsymbol{\theta}_i \in \Theta$ **do**
12: $\quad\quad\quad f_i' \leftarrow$ LOSS$(\boldsymbol{\theta}_i, \mathcal{D}, t), \Delta f_i \leftarrow f - f_i'$
13: $\quad\quad$ **end for**
14: $\quad\quad (f, \Delta f, \boldsymbol{\theta}) \leftarrow$ MIN$(\Delta f_i, f_i, \Theta)$ $\quad\quad\quad\quad\quad\quad\quad$ ▷ choose the optimal $\boldsymbol{\theta}$
15: $\quad$ **end while**
16: $\quad$ **return** $\boldsymbol{\theta}$
17: **end procedure**

---

## 3.1 SA-SHADE-TRI-ENSIN

The performance of DE relies on a proper selection of mutation strategy, scaling factor $F$, and crossover rate $Cr$ (See Supplementary Material Section S2.3 (sup)). It can be prohibitively expensive to hypertune them for training DNNs, especially when training very deep DNNs on large-scale datasets. Adaptive DE, where the strategy or the two parameters are self-adapted to the search experience, is beneficial for selecting the strategies and adjusting the parameters during the evolutionary optimization. Two powerful variants of DE were developed, namely Self-Adaptive Differential Evolution (SADE) (Qin & Suganthan, 2005) and Success-History based Adaptive DE (SHADE) (Tanabe & Fukunaga, 2013).

SADE made three main changes to improve the performance of DE. First, SADE records operations that produce better offspring. Second, SADE uses an adaptive crossover rate, $Cr$. The $Cr$ is drawn from the normal distribution, with the mean being the average of the previous $Cr$ that produces better offspring. Second, SADE uses the following four strategies $S_1$ - $S_4$ to perform mutation

$$\begin{aligned} S_1 : & \quad \theta_i^* = \theta_{r_1} + F \cdot (\theta_{r_2} - \theta_{r_3}) \\ S_2 : & \quad \theta_i^* = \theta_i + F \cdot (\theta_{gbest} - \theta_i) + F \cdot (\theta_{r_2} - \theta_{r_3}) \\ S_3 : & \quad \theta_i^* = \theta_{r_1} + F \cdot (\theta_{r_2} - \theta_{r_3}) + F \cdot (\theta_{r_4} - \theta_{r_5}) \\ S_4 : & \quad \theta_i^* = \theta_i + \text{rand}_u(0,1) \cdot (\theta_{r_1} - \theta_i) + F \cdot (\theta_{r_2} - \theta_{r_3}), \end{aligned}$$

where $i = 1, ..., ps$ and $r_1, r_2, r_3, r_4, r_5$ are random integers less than $ps$, different from $i$ and other. $gbest$ is the index of the best individual. Each strategy $S$ is chosen according to its probability of generating better offspring in history $P_S$, which is recorded with a queue of size $H$.

Another important DE variant is SHADE. It further enhances the performance of DE in optimization by updating the crossover rate and mutation rate in a more delicate manner. The weighted Lehmer mean is computed in every iteration. Using $F$ as an example, $\text{mean}_{WL}(F) = \sum_{i=k}^{n_s} w_k \cdot F_k^2 / \sum_{i=k}^{n_s} w_k \cdot F_k$, where $n_s$ is the number of successful update in the generation, $w_i = \Delta\mathcal{L}_i / \sum_{k=1}^{n_s} \Delta\mathcal{L}_k$ and $\Delta\mathcal{L}_k = |\mathcal{L}(\boldsymbol{\theta}_k^*) - \mathcal{L}(\boldsymbol{\theta}_k)|$. The values are then saved in the history archive with size $H$. In each generation, the $F$, $Cr$, and the frequency of sinusoidal function are generated by randomly selecting among the record and picking from the normal distribution of the randomly selected element $r_i$ with the archived value $M_{CR/F}$ ($\text{randn}(M_{CR/F,r_i}, V_{CR/F})$, where $V_{CR/F}$ is a hyper-parameter that defines the variance of distribution). For the neural network optimization, we did not use the Gaussian walk and the single strategy for a mutant generation. We use the four strategies from SADE and the historical archive of hyperparameters for each strategy. This cultivates a very rich variety of strategies when optimizing the neural network weight.

## 3.2 Tricks on mutation operation

We also adopt additional tricks to enhance the exploration and exploration capability of the optimizer by the trigonometric mutation (Fan & Lampinen, 2003) and the sinusoidal mutation rate (Awad et al., 2016). The trigonometric mutation operation, which has a strong local search capability, is performed according to, $S_{Trigo}$: $\theta_i^* = (\theta_{r_1} + \theta_{r_2} + \theta_{r_3})/3 + (p_2 - p_1)(\theta_{r_1} - \theta_{r_2}) + (p_3 - p_2)(\theta_{r_2} - \theta_{r_3}) + (p_1 - p_3)(\theta_{r_3} - \theta_{r_1})$, where $p_i = |\mathcal{L}(\theta_{r_i})|/p'$ and $p' = |\mathcal{L}(\theta_{r_1})| + |\mathcal{L}(\theta_{r_2})| + |\mathcal{L}(\theta_{r_3})||$.

The sinusoidal mutation rate provides a variety of scaling factors $F$ for exploration and exploitation. At the first half of the iterations in each batch (stage 1 $s_1$), where $g_{s_1} \in \left[1, \frac{G_{\max}}{2}\right]$, two different sinusoidal configurations are used, the first one is Non-Adaptive Sinusoidal Decreasing Adjustment in the left equation, and the second one is Adaptive Sinusoidal Increasing Adjustment in the right equation:

$$F_{i,g_{s_1}} = \frac{1}{2}\left(\sin\left(2\pi\,\nu\,g_{s_1} + \pi\right)\frac{G_{\max} - g_{s_1}}{G_{\max}} + 1\right), F_{i,g_{s_1}} = \frac{1}{2}\left(\sin\left(2\pi\nu_{i,k} * g_{s_1}\right)\frac{g_{s_1}}{G_{\max}} + 1\right),$$

where $\nu$ is the non-adaptive frequency, and $\nu_{i,k}$ is the adaptive frequency for strategy $k$ and individual index $i$.

## 3.3 Tricks on Population Initialization

Population initialization strategies determine the quality of the primordial soup before evolution. Besides the initialization using ending BP epochs, we also incorporate two other tricks in ending BP epochs, such as different random seeds (Init-RS), data augmentation techniques, and distinct hyperparameter (Init-HP) settings using TIMM (Wightman, 2019), which include simple and data-agnostic data augmentation routine, mixup, introduced by (Huang et al., 2017). For the hyperparameters, we adjusted the learning rate and weight decay. Additionally, we applied the smoothed loss function (Berrada et al., 2018) on the training process. The smoothed loss function is a regularization technique that aims to make the training process more stable by reducing the sensitivity of the model to outliers or noisy data points.

The overarching aim of these tricks was to increase the diversity among individual models and assess how these variations affect the model's overall performance. By introducing 'mixup' and 'smoothing' alongside hyperparameter adjustments, we aimed to comprehensively explore the potential enhancements in model performance across various settings.

In this paper, we hybridize the SADE and SHADE, with trigonometric mutation and the sinusoidal mutation rate. The proposed method is called SA-SHADE-tri-ensin. The method adapts its $F$, and $Cr$, and mutation strategies ($S_1$, $S_2$, $S_3$, $S_4$, $S_{Trigo}$) with probability of selection ($P_{S_1}$, $P_{S_2}$, $P_{S_3}$, $P_{S_4}$, $P_{S_{Trigo}}$) similar to Section 3.1, while evolving DNNs.

## 4 Experiment

### 4.1 Implementation

We utilize the CUDA/PyTorch framework to train a deep neural network using an evolutionary algorithm across 8 NVIDIA RTX A6000 GPUs. In comparison to BP with identical batch size and

network architecture, our empirical findings indicate a nearly halved GPU memory consumption, as gradients need not be recorded. Consequently, this affords us the ability to employ a larger batch size while maintaining the same graphical memory usage.

## 4.2 DATASETS AND PRE-TRAINED MODELS

Prior to the start of evolution, we used models with publicly available pre-trained weights for ImageNet (i.e., ResNet-18, ResNet-34, ResNet-50, and ResNet-101). The pretrained model is a vanilla version provided by the PyTorch torchvision library. We ensured consistency with the original method in terms of data, training strategies, and other aspects during the optimization process in DE. In other words, we did not employ any additional tricks such as auto data augmentation, label smoothing, dropout, etc during the evolutionary process. Additional experiments on smaller datasets and models are provided Section S2 in the supplementary (sup).

We optimized the pretrained models for $ps$ partial epochs using BP to initialize the population of DNNs for our proposed SA-SHADE-tri-ensin. The proposed SA-SHADE-tri-ensin evolves ResNet variants on ImageNet using the hyperparameters provided in Table S3 in the supplementary (sup). Each batch of data is used to evolve DNNs over ten generations. During the evolution, we exempted the fully connected layers from the evolutionary update to facilitate efficient training (Hinton et al., 2015). For the batch normalization (BN) layer, we exempted the update of running mean and variance but allowed them to learn the distribution during the forward feeding process.

## 4.3 MAIN RESULTS

### 4.3.1 IMPROVED PERFORMANCE OF DNNS

In Fig. 2, it is evident that all four variants of ResNets demonstrate performance improvements when employing the evolution framework with our developed SA-SHADE-tri-ensin algorithm. Our findings indicate that the framework exhibits robustness across a wide range of network depths, from ResNet-18 with 11.7 million parameters to ResNet-101 with 44.5 million parameters. Notably, in comparison to a recent related work that does not utilize gradient details (Whitaker & Whitley, 2023), our approach outperforms their method in ResNet-18, achieving an accuracy of 70.042% as opposed to their 69.93% (as shown in Table 1). Our approach does not involve ensemble methods, problem decomposition, or introducing batch-wise evolution during training. To further benchmark the performance, we pick a few other well-known techniques, Weight Averaging (WA) (Izmailov et al., 2018), Fast Geometric Ensembling (Ensemble) (Garipov et al., 2018), and Model soups (Wortsman et al., 2022) as our baseline methods. The highest accuracy is typically achieved within the first ten batches, corresponding to approximately 100 generations evolved, across all four ResNet variations. Validation is conducted at the end of each batch. The convergence of loss and accuracy generally occurs within approximately 20 batches, beyond which no significant improvements are observed. The mean Euclidean distance is also shown to converge as shown in Fig. 3. However, it should be noted that our results could potentially improve with extended training duration, as our experiments were limited to only 20 batches, roughly equating to 40,000 images, in contrast to the vast amount of data present in the ImageNet dataset.

### 4.3.2 IMPACT OF POPULATION INITIALIZATION TRICKS

The upper part of Table 2 summarizes the impacts of population initialization strategies on different methods trained on ImageNet with ResNet-50. It is generally observed that the populations created by Init-RS and Init-HP have average accuracy and the best accuracy higher than the pre-trained model provided by PyTorch. Moreover, the initial population generated by Init-RS is superior to Init-HP.

The initial population of DNNs is then combined, merged, or evolved using different methods summarized in the lower part of Table 2. It is found that Init-HP is generally better than Init-RS, except for model soups and DE. SA-SHADE-tri-ensin with Init-HP obtained the best Top-1 accuracy. The accuracy curve of SA-SHADE-tri-ensin is provided in Figure 4, demonstrating even 10 generations of SA-SHADE-tri-ensin are sufficient for superior performance.

Table 1: **Test accuracy of ResNets on ImageNet.** Comparative analysis between pre-trained models by PyTorch, sparse mutation decomposition (SMD), weight averaging (WA), Ensemble, and ours, which does not utilize population initialization tricks.

| | Top1@ | | | | | Top5@ | | | | | Params |
|---|---|---|---|---|---|---|---|---|---|---|---|
| | Pytorch1 | SMD34 | WA15 | Ensemble9 | Ours | Pytorch1 | SMD34 | WA15 | Ensemble9 | Ours | |
| ResNet-18 | 69.76 | 69.93 | 70.00 | 70.02 | 70.04 | 89.08 | na | 89.31 | 89.32 | 89.35 | 21.8M |
| ResNet-34 | 73.31 | na | 73.47 | 73.49 | 73.51 | 91.42 | na | 91.51 | 91.53 | 91.57 | 21.8M |
| ResNet-50 | 76.13 | na | 76.56 | 76.63 | 76.64 | 92.86 | na | 93.19 | 93.23 | 93.23 | 25.6M |
| ResNet-101 | 77.37 | na | 77.62 | 77.63 | 77.70 | 93.55 | na | 93.71 | 93.71 | 93.76 | 44.5M |

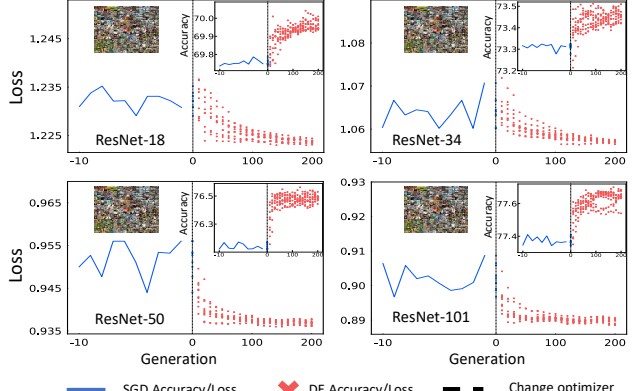

Figure 2: **Evolving ResNet Variants on ImageNet.** The test losses and accuracies for ResNet-18, ResNet-34, ResNet-50 and ResNet-101.

Figure 3: **Diversity and convergence.** Euclidean distance between DNNs.

| Population Initialization | Init-RS | | Init-HP | |
|---|---|---|---|---|
| | Top1 | Top5 | Top1 | Top5 |
| Pre-trained by PyTorch 1 | 76.13 | 92.86 | 76.13 | 92.86 |
| Initial Population (average) | 76.60 | 93.20 | 76.32 | 92.95 |
| Initial Population (best) | 76.62 | 93.22 | 76.48 | 93.00 |
| Weight Averaging 15 | 76.63 | 93.19 | 76.54 | 93.06 |
| Ensemble 9 | 76.61 | 93.19 | 76.65 | 93.14 |
| Model Soups 37 | 76.61 | 93.19 | 76.56 | 93.07 |
| DE | 76.63 | 93.21 | 76.58 | 93.09 |
| SA-SHADE-tri-ensin | 76.67 | 93.23 | 76.71 | 93.14 |

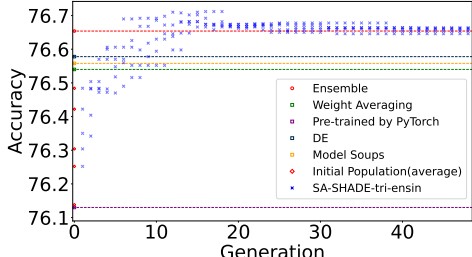

Table 2: **Impact of population initialization.** Comparative analysis of two population initialization tricks, Init-RS and Init-HP, on different methods, such as weight averaging, Ensemble, model soups, and ours trained on ImageNet with ResNet-50.

Figure 4: **Accuracy curve of SA-SHADE-tri-ensin.** Comparative analysis of SA-SHADE-tri-ensin trained on ImageNet with ResNet-50 with various methods, detailed in Table 2.

## 4.4 ABLATION STUDIES

### 4.4.1 IMPACT OF DIFFERENT SA-SHADE-TRI-ENSIN COMPONENTS

The proposed SA-SHADE-tri-ensin algorithm integrates various components from the state-of-the-art DE algorithms by hybridizing adaptive learning (i.e., SADE, SHADE) of mutation & recombination, trigonometric mutation, and sinusoidal change of mutation rate ($Sin$-$F$). We conducted experiments to verify the functionality of specific combinations of components by evolving ResNet-50 on ImageNet. The results are provided in Table. 3. According to the results, SA-SHADE-tri-ensin, which combined all components with Init-HP, performed the best.

Table 3: **Ablation study on different SA-SHADE-tri-ensin components.** The accuracy on ImageNet, when ResNet-50 is optimized using various combinations of SA-SHADE-tri-ensin components. Prior to evolving ResNet-50, the pre-trained models obtained a top-1 accuracy of 76.13%.

| DE | SADE | SHADE | Trigonometric | $Sin$-$F$ | Init-RS | Init-HP | Top-1 (%) |
|----|------|-------|---------------|-----------|---------|---------|-----------|
| ✓ | | | | | | | 76.421 |
| | ✓ | | | | | | 76.463 |
| | ✓ | ✓ | | | | | 76.506 |
| | ✓ | ✓ | ✓ | | | | 76.547 |
| | ✓ | ✓ | | ✓ | | | 76.522 |
| | ✓ | ✓ | ✓ | ✓ | | | **76.568** |
| | ✓ | ✓ | ✓ | ✓ | ✓ | | **76.664** |
| | ✓ | ✓ | ✓ | ✓ | | ✓ | **76.712** |

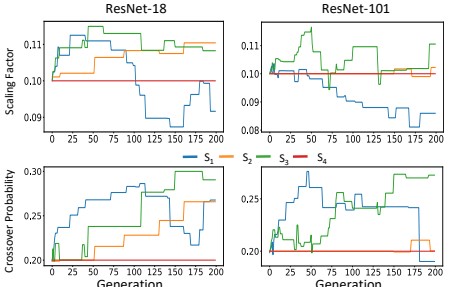

Figure 5: **The adaptation of $F$ and $Cr$.** Different $F$ and $Cr$ were selected to produce fitter offspring at different phases of the evolution for different strategies (colored lines) when evolving ResNet-18 and ResNet-101 on ImageNet.

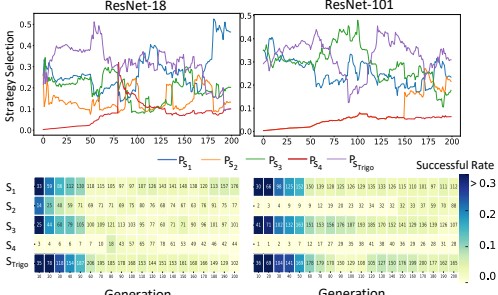

Figure 6: **Evolution dynamics.** Visualization of the competition and collaboration between mutation strategies when evolving ResNet-18 and ResNet-101 on ImageNet. (Upper) The probability of selecting the strategy. (Lower) The successful rate of the strategy.

### 4.4.2 IMPACT OF MUTATION AND CROSSOVER

Natural selection drives the evolutionary process by specifying the advantageous traits required for survival. Hence, species with better evolutionary operators (i.e., mutation and recombination) have competitive advantages in producing fitter offspring. We show in the Supplementary Material Section S2.3 (sup) that the enhanced performance only occurs when the mutation and recombination are appropriately chosen, enabling the inheritance of advantageous traits from the parents while maintaining diversity in the population; this supports the need for the adaptive feature in our proposed algorithm SA-SHADE-tri-ensin. We show in Fig. 5 the mean of the archived values for both the $F$ and $Cr$, which successfully produce better offspring. The mean of the archived value varies with generations depending on the update strategy. Generally, the scaling factor fluctuates slightly around 0.1, while the crossover rate increases throughout the generations. We note that both hyperparameters of the strategy $S_4$ merely have no change. Hence, it is not an inefficient mutation strategy for DNNs.

### 4.4.3 DIFFERENT DATASET AND DNNs

To examine the generalization capabilities of the proposed framework, the evolutionary processes of different DNNs on different datasets, mentioned in Sec. 4.2, were analyzed. Based on the results (detailed in Sec. S2 of the Supplementary Materials (sup)), the experiment showed that the proposed method leads to decreases in validation loss and increases in validation accuracy over generations. Compared to BP-based methods, the proposed framework shows enhanced classification performance; no overfitting problem is observed in the EA training, demonstrating advantageous effects similar to regularization; low time complexity (refer to Sec. S4 of the Supplementary Materials (sup)), which make it highly practical to be incorporated into the recent framework of DNN training.

### 4.4.4 HYPERPARAMETERS OF SA-SHADE-TRI-ENSIN

The impacts of hyperparameters and initialization method were analyzed in Sec. S3 of the Supplementary Materials (sup). It is observed that when the batch size of the proposed DE is larger than the batch size used by SGD, there will be a mismatch of values in the BN layer and cause deteriorated performance. Population size was observed to affect the converging behavior. Moreover, proper population initialization was found to be crucial to evolve DNNs in the original search space rather than the decomposed search space (Gong et al., 2020; Whitaker & Whitley, 2023).

### 4.4.5 THE ADAPTION OF MUTATION STRATEGIES

Fig. 5 and 6 show the adaptive behavior of the proposed SA-SHADE-tri-ensin, where mutation strategies, scaling factor, and crossover rate collaborate and compete to produce fitter offspring. In Fig. 5, it is observed that different $F$ and $Cr$ are selected to produce better offspring at different phases of evolution. In Fig. 6, the lower panel shows the success rate of the strategy with deeper color for a higher rate. The numbers in small boxes are the number of trials for certain strategies in the recent 50 generations. The strategies $S_1$, $S_3$, and $S_{Trigo}$ are observed to have a higher probability of being selected and produce better offspring with higher success rates than strategies $S_2$, $S_4$. The strategy $S_{Trigo}$, the only mutation strategy that utilizes the information of the loss function, turned out to be the dominant strategy, especially in the first 50 generations. As evolution progress, the loss function saturates, no matter which evolutionary operators are used.

### 4.5 DIVERSITY AND CONVERGENCE OF THE POPULATION

Darwin's principle of divergence proposed that species diversity might increase the productivity of ecosystems because of the division of labor among species. Hence, it is intriguing to analyze how DNNs' diversity manifests differently at various stages of the evolutionary process, which can be observed by comparing Figs. 2 and 3. As the evolution begins, a higher diversity is observed, together with stronger search capabilities, where better solutions are found over early generations. However, as evolution progresses, it is observed that diversity declines rapidly and stabilizes after approximately 50 generations. The development of a new strategy for further convergence is worth investigating.

## 5 CONCLUSIONS

Biological adaptation has historically emerged as a consequence of learning and evolution. In this study, we examine the effects of evolution on a population of DNNs derived from the ending epochs of BP with distinct hyper-parameters, drawing inspiration from Hinton's pretraining concepts (Hinton, 2022; Hinton & Salakhutdinov, 2006). To address the challenges associated with training deep neural networks and handling large-scale datasets such as ImageNet, we propose an adaptive DE algorithm, referred to as SA-SHADE-tri-ensin, which integrates state-of-the-art DE algorithms, namely SADE and SHADE, with trigonometric mutation and sinusoidal modulation of the mutation rate. Our proposed approach successfully improves the performance of all four ResNet variants. In comparison to other non-EA baselines and existing works (Whitaker & Whitley, 2023) that evolve DNNs without leveraging gradient details, our method achieves superior enhancement of pre-trained deep neural network models on the ImageNet dataset.

Our analysis reveals that the strategy of trigonometric mutation yields improved offspring with higher success rates and the important of diversity in parent population, particularly during the early stages of optimization. Remind that the "No Free Lunch theorem" (Wolpert & Macready, 1997) posits that there exists no universally superior optimization algorithm capable of outperforming others across all possible problems. Currently, neural network optimization remains in a black box state due to the extremely high dimensionality (Bai et al., 2021). This encourages the invention of novel strategies for further enhancements in neural network performance. While our adaptive DE algorithm effectively enhances existing deep ResNet variants on the ImageNet dataset, additional investigations are necessary to assess its performance in other DNN architectures and alternative learning tasks, such as image segmentation and image synthesis. We have also conducted a preliminary exploration of another significant evolutionary algorithm paradigm, particle swarm optimization (Kennedy & Eberhart, 1995), which demonstrates promising indications of performance improvement in the MNIST study (Refer to Sec. S5 of the Supplementary Materials (sup)).

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

# Supplemental Material for "Evolving Deep Neural Network's Weights at ImageNet Scale"

**Anonymous authors**

## S1 Correspondence between Darwinian Evolution and Neural Network Optimization Using Differential Evolution

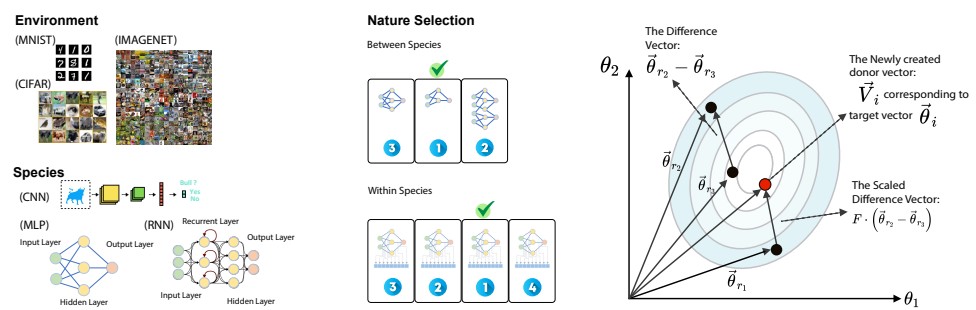

Figure S1: **Conceptual illustration of our proposed Darwinian evolution on neural networks using DE.** (Left) In analogy to Darwinian evolution, the dataset provides the environment where different types of neural networks strike to survive. (Middle) The evolution (natural selection and inheritance) applies to different network architectures and trainable weights in the same architect. Pretrained neural networks are the primordial ancestors for the DE to evolve and to select the 'elite' solution. (Right) An illustration of DE mutation.

The theory of evolution (Smith, 1993), supported by evidence from genetics, paleontology, and geology, describes how living beings originate from primordial soup, make up the first species (primordial ancestor, also known as last universal common ancestor (LUCA)), survive under environmental pressure, and evolve in nature over long periods. Since Charles Darwin's book "On the Origin of Species" was published in 1859 (Darwin, 2004), the theory has been expanding to describe the necessary conditions, ingredients, and mechanisms before the start of and the transition to evolution. In Fig. S1, we illustrate Darwinian Evolution's analogy to the neural network optimization problem using differential evolution (DE). The neural network architecture and the dataset play the role of the species and environment, respectively. Different architectures specialize in different functions, for example, convolutional neural networks (e.g., ResNet (He et al., 2015), MobileNet (Howard et al., 2017)) and recurrent neural networks (e.g., LSTM (Hochreiter & Schmidhuber, 1997), GRU (Chung et al., 2014)) capture translation invariances and temporal dependencies underlie the data by implementing the inner workings of the visual cortex and memory. The trainable parameters can be interpreted as the genetic traits within the architect framework that affects survival and breeding. On the other side, the complexity of the dataset can be interpreted as the complexity of the environment, scaling from simple MNIST(LeCun et al., 1998) to big data ImageNet(Deng et al., 2009). Survival fitness can be defined as the loss function of the learning tasks. This work focuses on ANN training by considering the evolution of a population of trainable parameters in the pre-defined architecture (i.e., single organism) rather than evolving a population of different architectures (Real et al., 2017; Lu et al., 2020).

A detailed description of DE is provided in the Method section of the main text. In Fig. S1, an illustration of the DE's mutation strategy $S_1$ in 2D search space is shown, with the contour lines indicating fitness functions in the search space. The figure shows how DE simulates mutation using any three candidate solutions in the population. Various improvements have been on the ordinary DE by adding adaptive learning (i.e., SADE, SHADE) of mutation & recombination, trigonometric and sinusoidal mutations ($Sin$-$F$) for scaling factor $F$.

### S1.1 Concept of Primordial soup and ancestor

Primordial soup theory (Taylor, 2005), the Miller-Urey experiment (Miller, 1953), and others (Kasting, 1993) studied and simulated the conditions of early Earth for the first life, which arose from non-living matters, give rise to other species through evolution. This work examined neuro-evolution that begins with the primordial ancestor, ADAM-trained neural networks as the first species. Neuro-evolution that starts from the primordial ancestor is found to be empirically superior to the primordial soup, randomly initialized neural networks, as shown in Fig. S2. Hence, the results align with the Primordial soup theory, showing the importance of forming a species before neuro-evolution takes place.

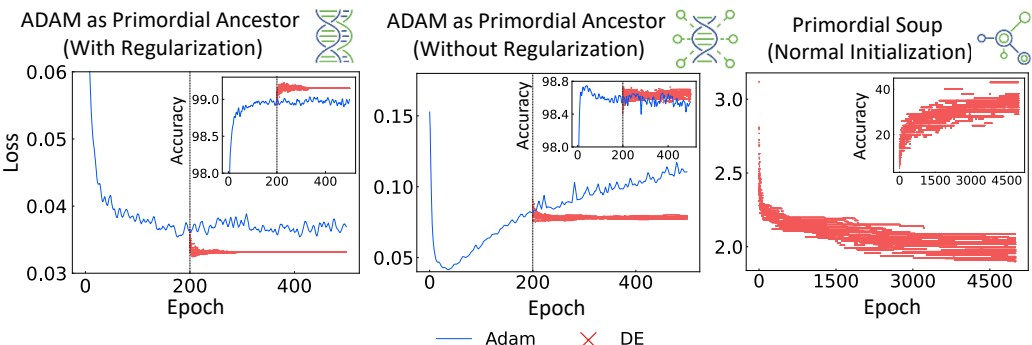

Figure S2: **The starting point of neuro-evolution: primordial ancestor v.s. primordial soup.** The left and middle figures illustrate how the losses and accuracies change over the epoch for DNNs trained by Adam (blue curves) and DE (red markers) using Adam as the primordial ancestor, with and without regularization. It is observed that Adam requires regularization to handle overfitting, while DE does not require. The right figure evolves a neural network from the primordial soup using random initialization, which has difficulty in convergence.

## S2 Small dataset and LeNets

We carried out experiments in small datasets, MNIST, Fashion MNIST, CIFAR-10, and CIFAR-100, and we trained models (i.e., LeNet1, LeNet5, MLP, RNN) from scratch using BP with early stopping and used the DNNs from the ending epochs (the last $ps$ epochs) as the initial population for ordinary DE. The hyperparameters of DE being examined are summarized in Table S2. The ordinary DE with strategy $S_1$ and crossover are used to evolve DNNs. The moderate size of datasets and DNNs allows us to examine different configurations of evolving DNNs in a controllable way.

In Fig.S3, LeNet1 and LeNet5 architectures are evolved in the environment (dataset) with different complexities, which provide different amounts of training data generated using data augmentation. The evolution of DE is observed to have positive impacts on the ADAM-trained neural network. Moreover, datasets and models with higher complexity are observed to achieve better performance. Both the LeNet1 and LeNet5 evolved using the proposed Darwinian evolution framework are found to be better than the LeNet5 trained using BP by LeCun et al. in 1998 (Lecun et al., 1998). The result is summarized in Table S1, with the error curve shown in Figure S4. The middle panel of Fig. S3 shows the degrees of improvement on top of the BP-based approach under different degrees of $L2$ regularization. It is observed that the best results are achieved with a regularization of 0.0001 compared to no regularization at all, and all regularization parameters lead to increased model accuracy. The use of regularization reduces the impact of over-fitting in BP-based optimizers. During the Darwinian evolution, it is observed that the trait of preventing over-fitting is inherited without an explicit $L2$ regularization in the fitness function. To assess the out-of-distribution robustness of the neural network trained using the proposed framework, two datasets with common corruptions are used, namely MNIST-C and CIFAR-10-C. The types of corruption are summarized at the right panel of Fig. S3.

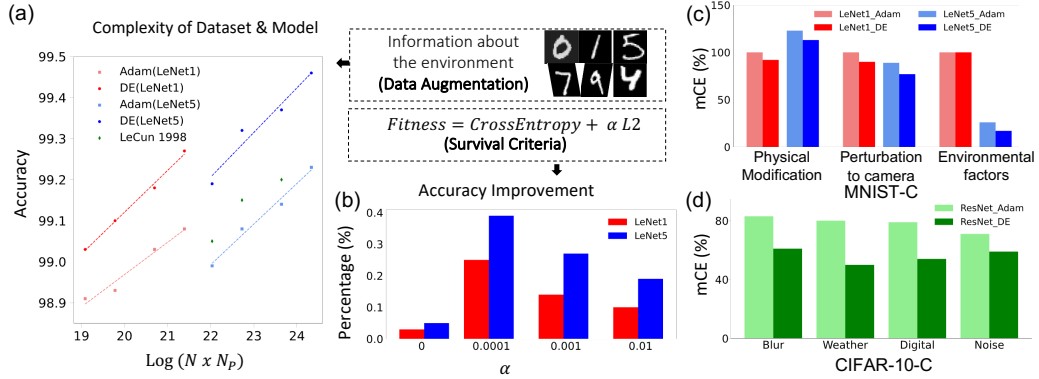

Figure S3: **The impact of environment (dataset and loss function) on the DNNs.** The left figure shows the relationships between the accuracy and the complexity of the dataset (MNIST with data augmentation) and model (LeNets). The middle bottom figure illustrates the influence of regularization. The two figures at the right show the performance of ADAM and DE on the corrupted MNIST-C and CIFAR-10-C.

Table S1: **Test accuracy of LeNets on MNIST and CIFAR.** (M) MNIST, (F.M) Fashion MNIST, (C) CIFAR

|  | LeCun's (Lecun et al., 1998) Acc | Adam Acc | DE Acc | Params |
|---|---|---|---|---|
| LeNet1(M) | 98.30 | 98.78 | 99.03 | 3,246 |
| LeNet5(M) | 99.05 | 98.95 | 99.20 | 62,006 |
| LeNet5(F.M) | na | 89.27 | 89.43 | 62,006 |
| LeNet5(C-10) | na | 60.16 | 61.36 | 62,006 |
| LeNet5(C-100) | na | 24.47 | 26.73 | 62,006 |

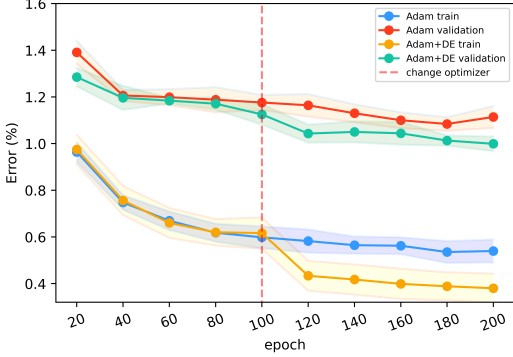

Figure S4: **Average classification error with standard errors across epochs during the training process.** LeNet-1 model consistently converges across 10 runs on MNIST.

## S2.1 DARWINIAN EVOLUTION DOES NOT OVERFIT

In nature, species evolve for better fit but not overfit (e.g., giraffes can reach higher leaves with the evolved long necks, but not over-long necks (Wang et al., 2022)). In this work, we studied whether

evolving neural networks using DE will lead to overfitting that gradient descent methods suffer. Generally, DE is found to have no overfitting issue, as shown in Fig. S2. Comparing the primordial ancestor trained with and without regularization, it is also observed that the quality of the primordial ancestor has a significant impact on DNNs, demonstrating the effect of descent in the same lineage. In the same lineage, offspring receive the genetic traits from parents through inheritance with slight variation and modification. In Fig. S2, the lineage trained using BP with regularization performs better, as the use of regularization reduces the impact of over-fitting in BP-based optimizers. During the Darwinian evolution, it is observed that the trait of preventing over-fitting is inherited without an explicit $L2$ regularization in the fitness function.

## S2.2 PERFORMANCE ON DIFFERENT DATASETS AND MODELS

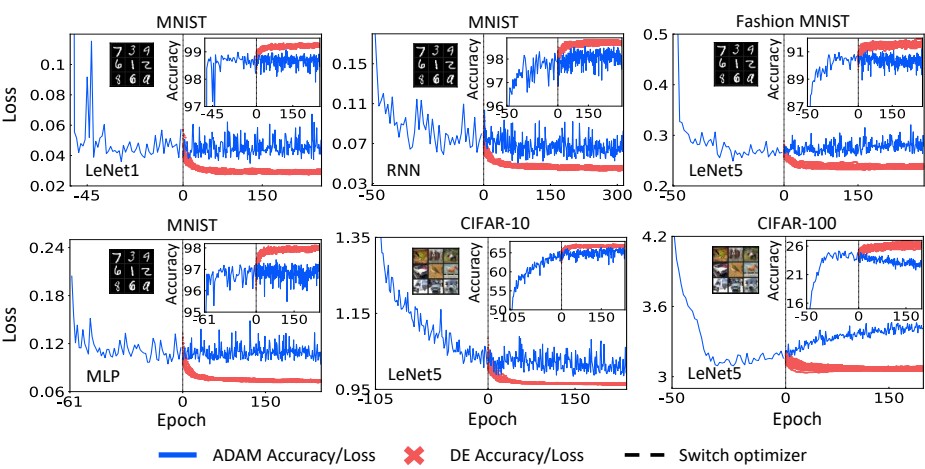

Figure S5: **Generalization of DE on different datasets and deep learning models.** Datasets include MNIST, Fashion MNIST, CIFAR-10, and CIFAR-100, while models include: LeNet1, LeNet5, MLP, and RNN. The blue lines represent Adam optimizer, and the red points represent the DE optimizer.

## S2.3 IMPACT OF SCALING FACTOR AND CROSSOVER RATE

Our comprehensive analysis has revealed the crucial significance of selecting appropriate parameters, namely $F$ and $Cr$, shown in Figs. S6 and S7. Hence, an adaptive approach is employed to enhance the performance of the DE optimizer to evolve DNN on ImageNet.

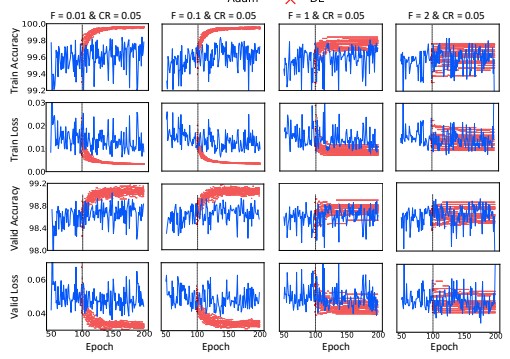

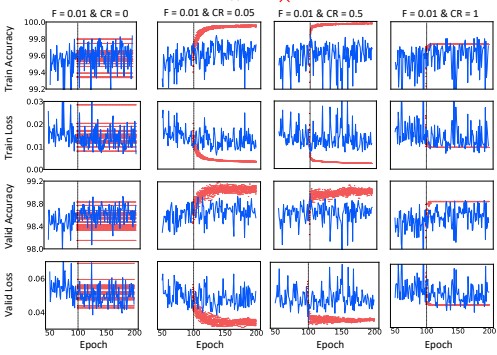

Figure S6: The impact of the mutation factor $F$. Comparison between DE (blue curves) and Adam (red marker) for RNN models, with fixed crossover rate and different F.

Figure S7: The impact of the crossover rate $CR$. Comparison between DE (blue curves) and Adam (red marker) for RNN models, with fixed mutation factor and different CR.

## S2.4 Hyperparamets Used

<table>
<tr><td colspan="3">Table S2: Hyper-parameters for DE.</td></tr>
<tr><td>Name</td><td>Symbol</td><td>Value/ Range</td></tr>
<tr><td>Scaling factor</td><td>$F$</td><td>[0.01, 0.1, 1, 2]</td></tr>
<tr><td>Crossover rate</td><td>$Cr$</td><td>[0, 0.05, 0.5, 1]</td></tr>
<tr><td>Mutation strategy</td><td>$S$</td><td>$S_1$</td></tr>
<tr><td># of generations</td><td>$Gen$</td><td>200-300</td></tr>
<tr><td>Batch size</td><td>$bs$</td><td>50000</td></tr>
<tr><td>Population size</td><td>$ps$</td><td>20</td></tr>
<tr><td>BP algorithm</td><td></td><td>ADAM</td></tr>
<tr><td>Update scope</td><td></td><td>whole NN</td></tr>
<tr><td>Regularization</td><td>$\alpha$</td><td>$[0, 10^{-4}, 10^{-3}, 10^{-2}]$</td></tr>
</table>

<table>
<tr><td colspan="3">Table S3: Hyper-parameters in SHADE-tri-ensin</td></tr>
<tr><td>Name</td><td>Symbol</td><td>Value/ Range</td></tr>
<tr><td>Scaling factor</td><td>$F$</td><td>0-0.2</td></tr>
<tr><td>Crossover rate</td><td>$Cr$</td><td>0-0.3</td></tr>
<tr><td>Mutation strategies</td><td>S</td><td>$S_1$ - $S_4$, $S_{Trigo}$</td></tr>
<tr><td># of generations</td><td>$Gen$</td><td>200</td></tr>
<tr><td>Batch size</td><td>$bs$</td><td>2048</td></tr>
<tr><td>Population size</td><td>$ps$</td><td>10</td></tr>
<tr><td>BP algorithm</td><td></td><td>SGD pretraining</td></tr>
<tr><td>Update scope</td><td></td><td>exempt FC</td></tr>
<tr><td>Archive size</td><td>$H$</td><td>5</td></tr>
</table>

## S2.5 Robustness against Noise

To verify the robustness and generalization of our approach, we trained the model using MNIST data and CIFAR-10 data and tested it on the MNIST-C and CIFAR10-C datasets, respectively. Table S4 shows the LeNet1 and LeNet5's performance on MNIST-C. Table S5 shows the LeNet1 and LeNet5's performance on CIFAR10-C. Table S6 shows the ResNet performance on CIFAR10-C.

Table S4: **Performance on MNIST-C.** The error and mean corruption error (mCE) of ADAM and DE for LeNet models trained on MNIST. Models that perform the best for each type of corruption is colored blue. Overall, LeNet5 trained by DE is found to be less susceptible to corruptions.

| | Error | | | | mCE | | | |
|---|---|---|---|---|---|---|---|---|
| | LeNet1_Adam | LeNet1_DE | LeNet5_Adam | LeNet5_DE | LeNet1_Adam | LeNet1_DE | LeNet5_Adam | LeNet5_DE |
| Shot Noise | 2.7% | 2.4% | 3.0% | **2.1%** | 100% | 92% | 114% | **79%** |
| Impulse Noise | 13.6% | 14.2% | 11.3% | **9.6%** | 100% | 104% | 83% | **70%** |
| Glass Blur | 13.1% | 12.2% | 8.9% | **6.3%** | 100% | 93% | 68% | **48%** |
| Motion Blur | 20.1% | 15.6% | 10.8% | **8.9%** | 100% | 78% | 54% | **44%** |
| Shear | 3.5% | 3.1% | 3.1% | **2.5%** | 100% | 89% | 88% | **74%** |
| Scale | 11.2% | **7.9%** | 12.4% | 8.1% | 100% | **71%** | 111% | 73% |
| Rotate | 9.6% | 8.7% | 9.5% | **7.9%** | 100% | 90% | 99% | **83%** |
| Brightness | 84.9% | 84.8% | 8.2% | **4.2%** | 100% | 100% | 10% | **5%** |
| Translate | 44.9% | 43.0% | 43.3% | **42.0%** | 100% | 96% | 96% | **94%** |
| Stripe | 32.2% | 31.8% | 15.1% | **8.2%** | 100% | 99% | 47% | **26%** |
| Fog | 81.9% | 82.4% | 22.0% | **18.2%** | 100% | 101% | 27% | **22%** |
| Spatter | 5.2% | 4.9% | 3.1% | **2.6%** | 100% | 95% | 59% | **49%** |
| Dotted Line | 4.8% | 4.6% | 4.0% | **3.6%** | 100% | 96% | 84% | **76%** |
| Zigzag | 16.0% | 15.4% | 13.0% | **11.3%** | 100% | 96% | 81% | **71%** |
| Canny Edges | 22.9% | **20.1%** | 40.2% | 37.9% | 100% | **88%** | 176% | 165% |
| Average | 24.4% | 23.4% | 13.9% | **11.6%** | 100% | 92% | 80% | **65%** |

Table S5: **Performance on CIFAR10-C.** The error and mean corruption error (mCE) of ADAM and DE for LeNet models trained on CIFAR10. Models that perform the best for each type of corruption is colored blue. In all corruptions, LeNet5 trained by DE is found to be less susceptible to corruptions.

| | Error | | | | mCE | | | |
|---|---|---|---|---|---|---|---|---|
| | LeNet1_Adam | LeNet1_DE | LeNet5_Adam | LeNet5_DE | LeNet1_Adam | LeNet1_DE | LeNet5_Adam | LeNet5_DE |
| Gaussian | 52.7% | 52.3% | 44.4% | **43.7%** | 100% | 99% | 84% | **83%** |
| Shot | 51.2% | 51.2% | 42.5% | **42.3%** | 100% | 100% | 83% | **83%** |
| Impulse | 56.2% | 56.1% | 48.3% | **47.4%** | 100% | 100% | 86% | **84%** |
| Defocus | 51.3% | 49.9% | 44.2% | **43.4%** | 100% | 97% | 86% | **85%** |
| Glass | 53.3% | 52.1% | 48.0% | **46.9%** | 100% | 98% | 90% | **88%** |
| Motion | 53.2% | 51.9% | 49.0% | **47.4%** | 100% | 97% | 92% | **89%** |
| Zoom | 53.5% | 52.1% | 48.5% | **46.9%** | 100% | 97% | 91% | **88%** |
| Snow | 52.1% | 52.1% | 46.4% | **46.0%** | 100% | 100% | 89% | **88%** |
| Frost | 54.9% | 55.9% | 52.4% | **50.2%** | 100% | 102% | 96% | **92%** |
| Fog | 58.8% | 57.6% | 56.7% | **54.7%** | 100% | 98% | 96% | **93%** |
| Brightness | 50.8% | 49.8% | 44.4% | **43.5%** | 100% | 98% | 87% | **86%** |
| Contrast | 67.2% | 65.5% | 65.3% | **64.0%** | 100% | 97% | 97% | **95%** |
| Elastic | 52.7% | 51.7% | 45.8% | **45.0%** | 100% | 98% | 87% | **85%** |
| Pixel | 49.7% | 49.1% | 41.7% | **41.1%** | 100% | 99% | 84% | **83%** |
| JPEG | 49.7% | 48.7% | 42.0% | **41.1%** | 100% | 98% | 84% | **83%** |
| Average | 53.8% | 53.1% | 48.0% | **46.9%** | 100% | 99% | 89% | **87%** |

Table S6: **Performance of deeper models on CIFAR10-C.** The error and mean corruption error (mCE) of ADAM and DE for deeper models trained on CIFAR10. Models that perform the best for each type of corruption is colored blue. Overall, deeper models trained by DE is found to be less susceptible to corruptions. ResNet and MobileNet are observed to be comparable, where each manages certain corruptions better.

| | Error | | | | mCE | | | |
| --- | --- | --- | --- | --- | --- | --- | --- | --- |
| | ResNet_Adam | ResNet_DE | MobileNet_Adam | MobileNet_DE | ResNet_Adam | ResNet_DE | MobileNet_Adam | MobileNet_DE |
| Gaussian | 37.5% | **30.9%** | 68.6% | 68.1% | 71% | **59%** | 130% | 129% |
| Shot | 36.0% | **28.4%** | 56.2% | 55.6% | 70% | **55%** | 110% | 109% |
| Impulse | 39.9% | **35.8%** | 49.2% | 49.1% | 71% | **64%** | 88% | 87% |
| Defocus | 40.3% | 26.9% | 21.7% | **21.2%** | 79% | 52% | 42% | **41%** |
| Glass | 42.4% | **37.8%** | 49.6% | 48.9% | 80% | **71%** | 93% | 92% |
| Motion | 49.3% | 33.8% | 31.1% | **30.3%** | 93% | 64% | 58% | **57%** |
| Zoom | 44.0% | 30.7% | 28.9% | **28.3%** | 82% | 57% | 54% | **53%** |
| Snow | 41.1% | 29.6% | 22.2% | **21.8%** | 79% | 57% | 43% | **42%** |
| Frost | 45.4% | 27.6% | 28.5% | **27.9%** | 83% | **50%** | 52% | 51% |
| Fog | 47.4% | 30.3% | 17.8% | **17.4%** | 81% | 51% | 30% | **30%** |
| Brightness | 39.5% | 21.4% | 9.3% | **9.2%** | 78% | 42% | 18% | **18%** |
| Contrast | 61.9% | 43.7% | 33.1% | **32.6%** | 92% | 65% | 49% | **49%** |
| Elastic | 41.6% | 28.0% | 20.5% | **20.0%** | 79% | 53% | 39% | **38%** |
| Pixel | 36.7% | **24.7%** | 26.4% | 26.9% | 74% | **50%** | 53% | 54% |
| JPEG | 36.0% | **23.0%** | 24.0% | 24.0% | 72% | **46%** | 48% | 48% |
| Average | 42.6% | **30.2%** | 32.5% | 32.1% | 79% | **56%** | 61% | 60% |

## S3   ADDITIONAL RESNETS EXPERIMENT ON IMAGENET

### S3.1   IMPACT OF POPULATION SIZE AND BATCH SIZE

Our method is similar to ensemble learning, where typically a larger number of models leads to better performance. Therefore, we increased the number of populations for analysis, as shown in Table S7. Additionally, in evolutionary algorithms, we require a certain number of samples to calculate fitness values for population selection. The mini-batch size of samples also has an impact on the evolution results, as demonstrated in Table S8.

Table S7: **Impact of population size.** The performance of different population size for evolution.

| Population size | 6 | 10 | 20 | 40 |
| --- | --- | --- | --- | --- |
| Top-1 (%) | 76.542 | 76.568 | 76.611 | **76.648** |

Table S8: **Impact of batch size.** The mini-batch samples for computing fitness value with DE algorithms.

| DE batch size | 256 | 512 | 1024 | 64000 | 320000 | 980000 |
| --- | --- | --- | --- | --- | --- | --- |
| Top-1 (%) | 76.542 | **76.568** | 76.553 | 76.551 | 76.561 | 76.563 |

### S3.2   POPULATION INITIALIZATION

In this paper, the population initialization is obtained by fine-tuning a pre-trained model. To validate the importance of this approach, we replaced the fine-tuning method with randomly generated Gaussian white noise for population initialization, i.e. Ref. (Whitaker & Whitley, 2023). Specifically, we computed the standard deviation of the initially fine-tuned population and used this standard deviation to generate Gaussian noise, which was then added to the pre-trained model. This resulted in a randomly initialized population. From the Table S9, it can be observed that the random noise initialization did not provide any benefits to our method.

## S4   TIME COMPLEXITY

Remarkably, if a $m$-layers network is employed, $\sum_{i=1}^{m} l_i$ parameters are to be optimized, where $l_i$ represents number of $i$-th layer. Time complexity are thus $\mathcal{O}(n_g \cdot \prod_{i=1}^{m-1} l_i l_{i+1})$ and $\mathcal{O}(n_e \cdot$

Table S9: **Comparison with randomly initialized population.**

| Method | Pytorch Benchmark | Random noise | Fine-tuned parents |
|---|---|---|---|
| Top-1 (%) | 76.13 | 76.15 | **76.57** |

$\sum_{i=1}^{m} l_i$) where $n_g$ and $n_e$ are training samples for gradient-based back-propagation and evolutionary algorithms, respectively.

Step by step, we analyze the algorithmic complexity for both gradient-based back propagation and evolutionary algorithm to train a layered neural network.

We emphasise that our analysis is on training a $m$-layered neural network, where $\sum_{i=1}^{m} l_i$ parameters are to be optimized. The procedure to produce update parameters are focused.

For feed-forward pass direction, each layer has experienced such process

$$Z_{i+1} \leftarrow M_{i+1,i} \cdot Z_i, \quad Z_{i+1} \leftarrow f(Z_{i+1}), \tag{S1}$$

where $f(*)$ is the activation function and $M_{i+1,i}$ contains the weights going from layer $i$ to $i+1$. Thus, time complexity is the same as feed-forward case, which in total demands $\mathcal{O}(n_g \sum_{i=1}^{m-1} l_i l_{i+1})$ basic operations and $\mathcal{O}(n_g \sum_{i=1}^{m-1} l_{i+1})$ queries to the inverse activation function.

For back-propagation direction, each layer has experienced such process

$$E_i \leftarrow f'(Z_i - O_i), \quad , D_{i,i-1} \leftarrow E_i \cdot Z_{i-1}, \quad M_{i,i-1} \leftarrow M_{i,i-1} - D_{i,i-1} \tag{S2}$$

where $E_{i-1}$ and $D_{i,i-1}$ are the error terms and adjust matrix. Remarkably, different algorithms are supposed to be employed here and we consider the typical case. Thus, time complexity is $\mathcal{O}(n_g l_i l_{i+1})$ operations, and $\mathcal{O}(n_g l_i)$ queries to the inverse activation function. In total, feed-forward pass algorithm demands $\mathcal{O}(n_g \sum_{i=1}^{m-1} l_i l_{i+1})$ basic operations and $\mathcal{O}(n_g \sum_{i=1}^{m-1} l_{i+1})$ queries to the activation function.

For the differential evolution algorithm, as described by equations in the Method section of the main text, demands $\mathcal{O}(n_e \sum_{i=1}^{m-1} l_i)$, including both the mutation operation and the crossover operation.

## S5 PARTICLE SWARM OPTIMIZER

To demonstrate that Darwinian evolutionary theory applies to a wide range of evolutionary algorithms, and the DE algorithm is not a special case. We selected the PSO algorithm (Kennedy & Eberhart, 1995) to replace DE and tested its optimization performance. As shown in Fig. S8, we find that the results for PSO are similar to those for DE in Fig. S2. It further proves the generalization and validity of our theory.

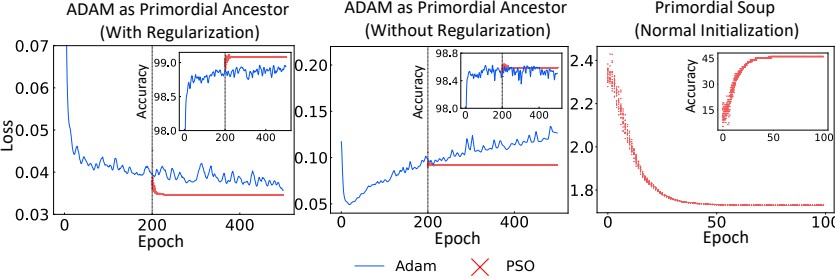

Figure S8: **Other nature-inspired optimizer** To demonstrate the generalization of our proposed method beyond differential evolution, another population-based optimizer, PSO, is used. Similar results in Figure 2 are observed. Hence, it further demonstrates the generalization and validity of the positive impacts of population-based optimizers on deep neural networks.

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
