# OpenReview forum: "Evolving Neural Network's Weights at Imagenet Scale"
_ICLR.cc/2024/Conference — Submitted to ICLR 2024_

### Official Review · Reviewer_SWqn · 2023-10-25

**Soundness:** 3 good
**Presentation:** 2 fair
**Contribution:** 1 poor
**Rating:** 3
**Confidence:** 4

**Summary:**

The paper argues that Evolutionary Algorithms (EA) can be used to optimize large deep nets, "ImageNet scale", such as ResNets, and be competitive against standard backpropagation/stochastic gradient descent (BP/SGD) methods, as long as EA methods are utilized only for fine-tuning previously pretrained nets with BP/SGD. The authors propose a differential evolution (DE)  based algorithm enhanced with several improvements proposed in DE literature over the years. With this algorithm, called SA-SHADE-tri-ensin, they perform an extensive battery of tests and ablation studies that provide evidence EAs, in specific, DE, can be used to train, or at least, fine-tune, deep neural nets (DNN).

**Strengths:**

- The paper is relatively well written. The main ideas of the authors are easy to follow, the authors take the liberty of introducing with enough detail required concepts and previous algorithms that are needed to understand their work.
- Authors provide extensive battery of tests that support their main idea, that training DNNs with EA is feasible up to a certain degree. Their experimental assessments include the proper usage of baseline methods.
- The paper revolves around an old intriguing topic, that is, if, or _how_, EAs can compete with BP/SGD. The paper itself is a contribution to this research topic, specially considering authors train very large, relatively modern, deep nets.

**Weaknesses:**

- The results show marginal improvements over vanilla methods (DE), less than 1% in accuracy improvement in ImageNet. Maybe authors could make emphasis in what version of ImageNet they're using, how many more images were accurately classified with their proposed method over previous approaches to better appreciate their results.
- Comparison with BD/SGD on a computational cost basis (hardware-time-score) is missing. This is fundamental to sustain the main claim of the article.
- Their proposed method looks like a mixed bag of tricks for enhancing DE, that authors combined hoping for the best. There is no section in the paper that argues, in a scientific or philosophical way, why they suspected their method should or would work. In this aspect, their paper falls short for a scientific article.
- In fact, if their method is only a combination of previously proposed DE enhancements, then there is no, or very little, contribution. This could be salvageable if accuracy improvements were to be dramatic, which they're not.

**Questions:**

I consider ICLR a top CS conference, and I expect submitted works to present radically novel and interesting ideas (even if they don't beat state of the art scores), or simple ideas (if they beat benchmarks in a dramatic way). I consider this paper to be none of the two cases: its contribution, from a scientific or technical point of view, is scarce, and quantitative results are negligible too (or at least, those that are presented with more emphasis).

I'd like to ask authors to rebate my point of view by providing some discussion and answers regarding two aspects of their work that could convince me otherwise:

- Hardware requirements. You briefly claimed that your proposed approach can alleviate hardware requirements in a dramatic way (nearly halving GPU memory), in comparison with BP/SGD. Could you provide a bit more detail on this? This could be the biggest selling point of your method. Democratizing AI methods by lowering hardware entry barriers is a big deal if you ask me, and it's something you should definitively make emphasis on. If you ask me, I would rewrite the whole paper by providing benchmarks on how hardware requirements are dramatically lowered, while just casually mentioning that scores are only marginally affected.

- Could you provide more details on the ImageNet version used? or actual numbers on how many more samples over the total size of the validation or test set are accurately classified on your ablation study? Because less than 1% over baseline methods frankly looks discouraging.

On an extra note, I'm also under the impression that authors actually wanted to explore the idea of using gradient methods as population initializing methods for EAs, which is very interesting if you ask me. Reviewing the supplementary material I got more convinced of this. Then, it looks like authors formatted their paper to fit within the realm of ICLR, talking about large deep nets, but providing poor benchmarking results. If I'm correct, then I'd suggest authors to reformat their paper for their original vision, even if it doesn't exactly fits ICLR scope. There are excellent EC venues too, which this paper could make a big contribution to, if properly re-envisioned.

---

### Official Review · Reviewer_4ELS · 2023-10-30

**Soundness:** 2 fair
**Presentation:** 2 fair
**Contribution:** 1 poor
**Rating:** 3
**Confidence:** 4

**Summary:**

This paper proposed to evolve the weights of DNNs at the ImageNet scale, which is an interesting topic because BP has been used for DNN training for many years with some limitations. The proposed work is implemented with DEs and some experiments are performed for the demonstration.

**Strengths:**

Using EAs to optimize weights at ImageNet scale.

**Weaknesses:**

The algorithm is enriched with tricks and also has no clear contribution against existing works. The experiments are toy ones, which could not demonstrate their usefulness.

**Questions:**

Using EAs to fine-tune the pre-trained model to enhance performance is not a new idea, but in this paper, I also did not see a significantly new aspect, not only the algorithm part but also the experiment results.

The algorithm parts, as highlighted by the section titles of sections 3.2 and 3.3, are enriched by different tricks.

A possible way to make the proposed work contribute to the community is to perform experiments for stat-of-the-art DNNs on ImageNet to verify if the proposed method can improve their reported performance, instead of a simple baseline.

The title of this paper overclaims what has been done in this paper, which easily misleads the readers that this work evolves the weights at ImageNet scale from statch.

Considering there are many popular EAs, what is the motivation for using DEs?

---

### Official Review · Reviewer_BX5K · 2023-10-31

**Soundness:** 2 fair
**Presentation:** 1 poor
**Contribution:** 1 poor
**Rating:** 3
**Confidence:** 5

**Summary:**

This article presents a neuroevolutionary method for weight optimization. Using Differential Evolution, weight updates from backpropagation are combined iteratively. This results in minor improvements to generalization compared to SGD, as demonstrated on MNIST, CIFAR-10, and Imagenet. The proposed approach, SA-SHADE-tri-ensin, is comparable to other neural recombination methods such as Fast Geometric Ensembling on Imagenet fine-tuning. Finally, the authors study various recombination methods and schedules for Differential Evolution.

**Strengths:**

The article makes a good argument for iteratively using ensemble methods such as Weight Averaging (WA) and Sparse Mutation Decomposition (SMD). S2.2 is, in my opinion, the most convincing result towards this point. It demonstrates that, at a point in SGD optimization, the use of an ensemble approach can continue improvement on the training set where SGD would otherwise stagnate or overfit. WA and SMD have been studied in the context of a single recombination of an ensemble or population of networks, while this article studies the iterative application of network recombination.

**Weaknesses:**

The main weakness of this article is in its proposed focus: optimizing networks once they have already been trained to a certain point using SGD. In other words, fine-tuning on the training dataset. While the results do show minor improvements over the base models, they are fractions of percentages on Imagenet. The use of a stochastic method (differential evolution) which is sensitive to hyperparameters (population size and crossover probability, in particular) suggests that these results may not even be reliable. There are no statistical tests which can attest to the performance of the proposed algorithm, so its value is difficult to determine.

This is separate from the motivation of articles used as comparison baselines for the proposed method, SA-SHADE-tri-ensin. SMD is motivated as an exploration of low-dimensional variation operators for evolutionary algorithms. Fast Geometric Ensembling (FGE) is proposed as a way to connect the loss basins of different models. WA is intended as a way to better use ensembles of models trained with different hyperparameters. I disagree with the authors that their approach does not involve ensemble methods, as an ensemble (population) of networks are used in recombination at each generation. As an ensemble method then, what is the added value of SA-SHADE-tri-ensin?

As I mention above, in my view, the added value is that it is studied iteratively. However, this is not very present in the article. Baseline methods like SMD, WA, and FGE do not appear to be studied iteratively, and training curves such as Figure 2 and Figure S5 only compare SGD with SA-SHADE-tri-ensin. If, as I see it, the main added value in this article is the use of ensemble averaging iteratively over training, this should be studied in detail.

The presentation of the article is also a weakness, which I expand upon below.

There are English errors or inaccuracies throughout the article. Many of the English errors are due to a misuse of articles. [This page](https://owl.purdue.edu/owl/general_writing/grammar/using_articles.html) might help the authors understand which article (a/an/the or no article) is appropriate.

There are multiple formatting errors in the references, such as missing journal and the listing of other authors as "Others". The arxiv version of published works is used on a few occasions, for example:
+ Whitaker, Tim and Darrell Whitley. "Sparse Mutation Decompositions: Fine Tuning Deep Neural Networks with Subspace Evolution." *Proceedings of the Companion Conference on Genetic and Evolutionary Computation (GECCO '23 Companion).* 2023.
+ Izmailov, Pavel, Dmitrii Podoprikhin, T. Garipov, Dmitry P. Vetrov and Andrew Gordon Wilson. “Averaging Weights Leads to Wider Optima and Better Generalization.” *Conference on Uncertainty in Artificial Intelligence.* 2018.

Backpropagation and stochastic gradient descent are conflated in the introduction. While the difference is clarified later, it should be clear throughout whether the author is referring to backpropagation or a gradient descent method like Adam.

The Related Works section could use better contextualization of the related works, rather than just listing them. Many sentences follow the format of "Author X did Y (cite)"; what about Y is interesting for inclusion in this article? Section 2.2 in particular suffers from this.

The notation throughout is unclear. In Equation 1, both normal θ and bold **θ** (\theta) are used, and in the algorithm, bold **θ** (\theta) appears to refer to an individual while capital (**Θ**) \Theta appears to refer to the population. Please see questions below for precise clarifications.

Finally, the biological metaphor which runs throughout the article, discussing the primordial soup and speciation, seems unnecessary. The algorithm being studied is an evolutionary algorithm, which is inspired by biology, however, is there specific pertinent motivation for the biological metaphors in this work? It is not about the study of the origin of life, or simulation of such, so the use of the primordial soup as an analogy falls flat. There is a single population which uses recombination between all individuals, so there is no concept of species in the algorithm. In S5, Differential Evolution is replaced with Particle Swarm Optimization and yet the concept of primordial soup is still present; these are two completely different metaphors. Metaphor and inspiration are laudable, but when used without motivation, they weaken scientific research.

**Questions:**

Was FGE applied over multiple epochs? It is in the original article.

Are $m$ and NP the same variable?

What is the difference between bold **θ** and normal θ?

What is the difference between lowercase θ and capital Θ?

Is $S\_{Trigo}$ a mutation or recombination operator?

Is there a mutation operator which only modifies a single individual, or is the mutation also a recombination?

What is $G_\{max}$?

---

### Meta-Review · Area_Chair_sQgh · 2023-12-11

**Metareview:**

In this paper, the authors present an evolutionary method for optimizing the weights of a deep neural network. The article presents their work on the image classification task using various convolutional neural network architectures. All reviewers commented positively on the clarity of exposition and the battery of tests employed by the authors. However, the reviewers also identified serious problems in terms of the marginal improvements reported, comparisons with standard baseline with SGD, and a lack of a theoretical justification for the methods. The authors did not respond to these concerns, and hence these concerns remain. For all of these reasons, this paper will not be accepted at this conference.

**Justification For Why Not Higher Score:**

Poor experiments. Unaddressed reviewer comments.

**Justification For Why Not Lower Score:**

n/a

---

### Decision · Program_Chairs · 2024-01-16

Reject